# Primary liver cancer in Georgia: Seven years' experience following the launch of the Hepatitis C Elimination Program, 2015–2022

Anna Khoperia[1]*, Susan H. Gawel[2,3], Jenna Bedrava[2,3], Nazibrola Chitadze[1], Maia Butsashvili[4,5], Alan Landay[6], Geoff A. Beckett[7], Sophia Surguladze[8], Konstantine Kazanjan[1], Vladimer Getia[1], Maia Tsereteli[1], Maia Alkhazashvili[1], Gavin A. Cloherty[2,3], Mamuka Zakalashvili[9], Amiran Gamkrelidze[10], Francisco Averhoff[2,3]

1 National Center for Disease Control and Public Health, Tbilisi, Georgia, 2 Abbott Pandemic Defense Coalition, Abbott Park, Illinois, United States of America, 3 Abbott Diagnostics, Abbott Park, Illinois, United States of America, 4 Human Research Union, Tbilisi, Georgia, 5 Clinic Neolab, Tbilisi, Georgia, 6 The University of Texas Medical Branch, Galveston, Texas, United States of America, 7 Division of Viral Hepatitis, Centers for Disease Control and Prevention, Atlanta, United States of America, 8 Integral Global Health, Tbilisi, Georgia, 9 Clinic Mrcheveli, Tbilisi, Georgia, 10 School of Health Sciences, University of Georgia, Tbilisi, Georgia

* a.khoperia@ncdc.ge

## Abstract

Hepatitis C virus (HCV) infection is associated with hepatocellular carcinoma (HCC), the predominant form of primary liver cancer (PLC). In 2015, Georgia implemented the HCV Elimination Program to reduce high national prevalence. We analyzed linked national registry data to characterize primary liver cancer cases in Georgia and evaluate the burden and treatment patterns of HCV infection among affected individuals during the national HCV elimination program. Data from the Georgian Cancer Registry (GCR), the national Vital Registration System, and the HCV Elimination Program databases were linked to perform a descriptive analysis of adult PLC cases diagnosed between January 2015-December 2022 and examine trends in HCV infection and treatment among them. Of 1734 reported PLC cases, 72.0% were male with a median age of 63 years (IQR 56–70). Two-thirds presented with stage 3–4 liver cancer and 80.2% were deceased by the end of 2022. Screening for HCV among PLC cases rose from 14.0% in 2015 to 92.8% in 2022. Of 1389 tested, 785 (56.5%) were anti-HCV positive. Of these, 654 (83.3%) were confirmed to be HCV RNA or HCV-cAg positive, 472 (72.2%) of which had advanced liver fibrosis or cirrhosis. Among 654 HCV RNA or HCVcAg positive individuals, 495 (75.7%) initiated HCV treatment, with 355 (71.7%) treated more than one year before their PLC diagnosis. Our analysis supports the known association between PLC and HCV infection in Georgia. The high proportion of PLC cases receiving HCV treatment prior to their liver cancer diagnosis highlights the need to establish routine post-treatment follow-up for cirrhotic patients to facilitate earlier detection of PLC and improve outcomes.

**Data availability statement:** This study relied on linked, individual-level data from multiple national surveillance systems in Georgia, including the Georgian Cancer Registry, the national Vital Registration System, and the Hepatitis C Elimination Program databases. These data are government-owned and governed by national data protection legislation. Although identifiers were encrypted prior to analysis, the dataset contains detailed clinical, temporal, and mortality information derived from multiple national registries. The ethical approval granted by the Institutional Review Board of the National Center for Disease Control and Public Health (IRB #2023-043) authorized the use of these data for the purposes of this approved research but does not permit public release of individual-level data. In addition, national regulations prohibit public dissemination of registry-level data derived from state surveillance systems. Therefore, the data cannot be made publicly available. Requests for access may be considered by the National Center for Disease Control and Public Health and the Ministry of Health in accordance with national data governance procedures and subject to formal review and data use agreements. For more information, please contact the Institutional Review Board of the National Center for Disease Control and Public Health at irb.ncdc@ncdc.ge.

**Funding:** This publication has been produced during the corresponding author's fellowship from the joint TEPHINET/APDC (Abbott Pandemic Defense Coalition) Field Epidemiology Training Program. This publication was supported by a Memorandum of Agreement funded by Abbott. Its contents are solely the responsibility of the authors and do not necessarily represent the official views of Abbott, The Task Force for Global Health, Inc. or TEPHINET. The co-authors from Abbott participated as employees and shareholders of Abbott.

**Competing interests:** The authors have declared that no competing interests exist.

## Introduction

Primary liver cancer (PLC) is the sixth most common cancer worldwide with nearly a million new cases diagnosed annually [1]. With the number of people dying from liver cancer predicted to rise by 56.4% by 2040, the burden of liver cancer is expected to increase [2]. An estimated 50 million people are living with hepatitis C virus (HCV) [3], a known risk factor for development of hepatocellular carcinoma (HCC), the most common subtype of PLC [4–6]. While only 20% of the global liver cancer burden is linked to chronic HCV, the estimate exceeds 50% in countries with high HCV infection prevalence [1,7].

In Georgia, a country with a population of 3.7 million [8], a national serosurvey in 2015 estimated 5.4% of the adult population was infected with HCV [9]. These findings prompted the launch of a national HCV Elimination Program in 2015 [10] to reduce the prevalence of active HCV infection by providing free testing and treatment with direct-acting antivirals (DAAs) [11].

To support the program, two interlinked HCV databases were created in 2015 and 2017 respectively. These databases collect real-time data on HCV testing and treatment in Georgia [12]. By the end of 2022, Georgia's HCV screening database contained results from 2.3 million persons tested for anti-HCV antibodies [13], and the treatment database contained information on over 78,000 individuals referred for HCV treatment [13].

As of 2021, Georgia has seen a 67% decrease in active HCV infections since the initiation of the program [14]. The program's impact on liver cancer incidence has become a subject of growing interest. A case-control study assessing the HCV-attributable HCC burden in Georgia in 2015–2019 found that HCV was significantly associated with a higher risk of liver cancer and HCC development in the Georgian population [15].

Despite Georgia's substantial progress toward HCV elimination, the characteristics of primary liver cancer cases and their relationship to HCV infection and treatment in the era of national elimination program remain incompletely described. In particular, there is limited evidence on the burden of active and treated HCV infection among PLC patients and the timing of antiviral treatment relative to cancer diagnosis at the population level. Using linked data from the Georgian Cancer Registry, the national HCV screening and treatment databases, and the national vital registration system, this study aimed to characterize the epidemiologic and clinical profile of PLC cases in Georgia from 2015–2022, determine the prevalence of active and treated HCV infection among them and evaluate patterns of HCV testing, treatment uptake, and timing of treatment in relation to PLC diagnosis to provide important baseline evidence to inform liver cancer surveillance strategies and guide long-term management of patients following HCV treatment in the context of national elimination efforts.

## Materials and methods

### Study design

This retrospective descriptive analysis examined the characteristics of primary liver cancer cases in Georgia to quantify associations with HCV infection, treatment, and

cure. Data on primary liver cancer cases diagnosed between January 2015-December 2022 were extracted from the Georgian Cancer Registry (GCR) and merged with data from HCV screening and treatment registries using each patient's unique 11-digit national ID number. The data were extracted on 15/06/2023 and included information updated through December 31, 2022. Participants' ID numbers were encrypted prior to the analysis to ensure anonymity of protected health information. The study was approved by the Institutional Review Board (IRB) of the National Center for Disease Control and Public Health (NCDC) of Georgia (IRB # 2023−043).

The GCR was established in 2015 to collect demographic, clinical, and received services information for confirmed and suspected cancer cases classified by the 10th revision of the International Classification of Diseases (ICD-10) codes at the national level [16,17]. For each registered case, the GCR collects the data on confirmed cancer cases using 3rd revision of the International Classification of Disease for Oncology (ICD-O-3) coding, stage (according to Tumor Nodes Metastases (TNM) classification system), results of laboratory investigations used in diagnosis, assigned treatment, and outcomes [16].

The initial screening for HCV infection is conducted using a rapid HCV antibody (anti-HCV) assay that tests for past or present HCV infection. Results of screenings are entered into the nationwide HCV screening database (StopC). Those who screen positive are referred for further testing using a polymerase chain reaction (RNA PCR) or HCVcAg assay to confirm active HCV infection. Those that are diagnosed with active HCV infection are then enrolled for treatment [18].

The HCV treatment database "Elimination C" (ElimC) stores demographic and clinical information on individuals enrolled into the HCV treatment program. It also includes the results of repeated HCV RNA testing to assess the achievement of sustained virological response (SVR), or cure, 12–24 weeks after treatment completion. Additionally, the database records patients' Fibrosis-4 score (FIB4) and transient elastography (TE) results, both of which are used to determine the degree of liver fibrosis [18]. Using liver transaminase levels, platelet counts, and age, FIB4 classifies patients as high-risk ($FIB4 > 3.25$), intermediate risk ($1.45 \le FIB4 \le 3.25$), or low risk ($FIB4 < 1.45$) for advanced liver fibrosis [19]. Transient elastography is a non-invasive technology measuring liver stiffness and has higher specificity compared to the FIB4 index [20]. The scoring system of TE ranges from F0 to F4, scaling from normal to minimal fibrosis (F0-F1 by Metavir: ≤ 7.0 kilopascals (kPa)), moderate (F2: 7.1–9.5 kPa), severe scarring (F3: 9.6–14.5 kPa) and cirrhosis (F4: > 14.5 kPa) [21].

Mortality data used in the analysis was extracted from Georgia's Vital Registration System, [16] a database that documents all-cause deaths based on issued death certificate records.

**Participant selection.** Participants included in the study were adults ≥18 years of age listed in the GCR as having any primary liver cancer (ICD-10 codes C22.0–C22.9) diagnosed from January 2015 through December 2022. Individuals younger than 18 years of age were excluded from the analysis.

The decision to include all primary liver cancer cases, as opposed to only including HCC cases, was based on a small-scale study that reported inconsistencies between assigned ICD-10 codes and patient medical charts for liver cancer diagnoses in 2015−2016 in Georgia [22]. This research indicated that liver cancer is often diagnosed at the advanced stages of disease, at times when a biopsy is not practical or beneficial. Such patients are usually reported as poorly specified primary or secondary liver cancer cases, resulting in the underestimation of the number of HCC-specific cancer cases. Given the context of these diagnostic limitations of liver cancer in Georgia, to ensure that all possible HCC cases were captured, the decision was made to include all patients with PLC-related ICD-10 codes in the analysis, rather than focusing on those specific to HCC alone (C22.0) [15,22].

**Variables and measurement.** Demographic variables examined in the analysis included age, sex, and mortality status. Death was defined as a present death date between January 1st, 2015, and December 31st, 2022. Cancer-related variables examined included PLC type, stage, and basis of diagnosis. We defined HCC as the subset of liver cancer patients with ICD-10 of C22.0. Other subtypes of reported PLC cases included intrahepatic cholangiocarcinoma (ICC) (ICD-10 C22.1), rare types of liver cancers (ICD-10 C22.2-C22.4), and poorly specified cancer types (ICD-10 C22.7 and ICD-10 C22.9). The basis of liver cancer subtype diagnosis was divided into two categories: tissue-based diagnosis

(including histology, cytology, immunohistochemistry and immunophenotyping) and diagnosis based on other diagnostic criteria (including clinical assessment, imaging studies (e.g., ultrasound, CT, MRI), and specific cancer markers).

HCV-related variables of interest included anti-HCV and HCV RNA or HCVcAg testing results. HCV negative status was defined as either a) anti-HCV negative or b) anti-HCV positive but HCV RNA or HCVcAg negative. HCV positive status was defined as a recorded positive HCV RNA or HCVcAg result, indicative of an active HCV infection. Anti-HCV positive cases without HCV RNA or HCVcAg testing results were excluded from the analysis, as their HCV status could not be fully determined.

For subjects enrolled in the HCV treatment program, additional clinical variables included HCV treatment status (when applicable), SVR results (when applicable), level of liver fibrosis, presence of chronic hepatitis B (HBV) co-infection (based on hepatitis B surface antigen (HBsAg) testing), and self-reported history of human immunodeficiency virus (HIV) infection. Transient elastography (TE) score was given priority over FIB4 score when available to assess the level of liver fibrosis. We classified liver fibrosis into three categories: low (TE score F0-F2 or FIB4 score <1.45), high (TE score F3-F4, or FIB4 score >3.25), and grey zone (in the case of a missing elastography result, those with a FIB4 score between 1.45 and 3.25). SVR achievement was defined for only those who completed at least one round of treatment and was based on patients' latest round of treatment.

**Statistical methods.** Our analysis examined characteristics of primary liver cancer patients, focusing on comparisons among HCV status groups and mortality status groups. Descriptive statistics were used to summarize and compare the distributions of demographic and clinical variables among both of these factors. Welch two-sample t-tests and Pearson's Chi-squared tests were used in hypothesis tests of binary and multinomial variables, respectively. Univariable odds ratios (OR) were used to quantify associations among categorical variables. We assessed trends in cancer diagnoses and HCV screenings and corresponding results over time to better understand the baseline burden of liver cancer throughout the first seven years of the HCV elimination effort. We performed a subset analysis of HCV-positive PLC patients, summarizing the HCV treatment initiation, completion, and outcomes, as well as the temporal relationship among these events. P-values < 0.05 were considered statistically significant. All statistical analyses were performed using R statistical software version 4.3.0.

## Results

Between January 2015 and December 2022, out of the 1743 primary liver cancer cases registered in the Georgian Cancer Registry, 1734 (99.5%) met the inclusion criteria for analysis, with 9 patients aged <18 years excluded. The overall median age was 63 years (IQR 56–70) at the time of cancer diagnosis (Table 1).

The majority of the registered cases were male (1248 cases, or 72.0%). Female PLC cases were significantly older (median age 69 years) than their male counterparts (median age 61 years) at the time of cancer diagnosis (p-value < 0.001). The majority of cases, 1391 (80.2%), died during the study period (January 2015 to December 2022). Most cases, 1144 (66.0%), presented with stage 3–4 liver cancer (Table 1). Overall, 1144 (66.0%) of the 1734 total cases did not have a specific liver cancer subtype classification assigned in the GCR and only 419 (24.2%) were classified as hepatocellular carcinoma (HCC) (Table 1). The diagnosis of HCC was twice as likely to be made in males compared to females (OR = 2.04, 95% CI = [1.56,2.71]).

The data showed an overall increase in the number of reported PLC cases over the study period with the most cases reported in 2020, followed by a decrease in cases in subsequent years (Fig 1).

Screening for anti-HCV antibodies among PLC cases increased from 14.1% in 2015 to 92.8% in 2022. Overall, 1389 (80.1%) cases diagnosed with PLC during 2015–2022 were screened at least once for anti-HCV (Table 1 and Fig 2).

The majority of cases, 1165 (67.2%), were tested for anti-HCV antibodies prior to PLC diagnosis (Fig 3).

A higher proportion of males had available anti-HCV test results compared to females (82.7% and 73.5% of cases with recorded screening test results, respectively). Out of 1389 cases tested for anti-HCV, 785 (56.5%) were positive.

**Table 1. Characteristics of the registered primary liver cancer cases in Georgia, 2015-2022.**

| | HCV Status* | | | Mortality Status as of December 2022 | | Overall |
|---|---|---|---|---|---|---|
| | Positive N = 654[1] (37.5%) | Negative N = 665[1] (38.4%) | Missing N = 415[1] (23.9%) | Alive N = 343[1] (19.8%) | Deceased N = 1,391[1] (80.2%) | N = 1,734[1] |
| **Demographics and Comorbidities** | | | | | | |
| **Sex a** | | | | | | |
| Female | 80 (12.2%) | 261 (39.2%) | 145 (34.9%) | 93 (27.1%) | 393 (28.3%) | 486 (28.0%) |
| Male | 574 (87.8%) | 404 (60.8%) | 270 (65.1%) | 250 (72.9%) | 998 (71.7%) | 1,248 (72.0%) |
| **Age a,b** | 59 (54, 65) | 67 (59, 73) | 65 (58, 75) | 60 (54, 67) | 64 (57, 72) | 63 (56, 70) |
| **Age Group a,b** | | | | | | |
| 18-49 | 59 (9.0%) | 56 (8.4%) | 35 (8.4%) | 50 (14.6%) | 100 (7.2%) | 150 (8.7%) |
| 50-59 | 281 (43.0%) | 120 (18.0%) | 97 (23.4%) | 115 (33.5%) | 383 (27.5%) | 498 (28.7%) |
| 60-69 | 232 (35.5%) | 246 (37.0%) | 129 (31.1%) | 115 (33.5%) | 492 (35.4%) | 607 (35.0%) |
| 70+ | 82 (12.5%) | 243 (36.5%) | 154 (37.1%) | 63 (18.4%) | 416 (29.9%) | 479 (27.6%) |
| **History of HIV Positive** | | | | | | |
| Yes | 1 (0.2%) | 0 (0.0%) | 0 (0.0%) | 0 (0.0%) | 1 (0.1%) | 1 (0.1%) |
| No | 298 (45.6%) | 0 (0.0%) | 0 (0.0%) | 87 (25.4%) | 211 (15.2%) | 298 (17.2%) |
| Missing | 355 (54.3%) | 665 (100.0%) | 415 (100.0%) | 256 (74.6%) | 1,179 (84.8%) | 1,435 (82.8%) |
| **HBV Status** | | | | | | |
| HBsAg Positive | 13 (2.0%) | 0 (0.0%) | 0 (0.0%) | 6 (1.7%) | 7 (0.5%) | 13 (0.7%) |
| HBsAg Negative | 514 (78.6%) | 0 (0.0%) | 0 (0.0%) | 127 (37.0%) | 387 (27.8%) | 514 (29.6%) |
| Missing | 127 (19.4%) | 665 (100.0%) | 415 (100.0%) | 210 (61.2%) | 997 (71.7%) | 1,207 (69.6%) |
| **HCV Testing** | | | | | | |
| **Anti-HCV Result c** | | | | | | |
| Anti-HCV Positive | 651 (99.8%) | 62 (9.3%) | 72 (100.0%) | 172 (57.3%) | 613 (56.3%) | 785 (56.5%) |
| Anti-HCV Negative | 1 (0.2%) | 603 (90.7%) | 0 (0.0%) | 128 (42.7%) | 476 (43.7%) | 604 (43.5%) |
| Not Tested | 2 | 0 | 343 | 43 | 302 | 345 |
| **Viremia Testing Result c** | | | | | | |
| HCV RNA or HCVcAg Positive** | 654 (100.0%) | 0 (0.0%) | 0 (NA%) | 147 (87.5%) | 507 (92.5%) | 654 (91.3%) |
| HCV RNA or HCVcAg Negative | 0 (0.0%) | 62 (100.0%) | 0 (NA%) | 21 (12.5%) | 41 (7.5%) | 62 (8.7%) |
| Not Tested | 0 | 603 | 415 | 175 | 843 | 1,018 |
| **Primary Liver Cancer** | | | | | | |
| **Cancer Type 2, a,b** | | | | | | |
| Hepatocellular Carcinoma (HCC) | 217 (33.2%) | 125 (18.8%) | 77 (18.6%) | 127 (37.0%) | 292 (21.0%) | 419 (24.2%) |
| Intrahepatic Cholangiocarcinoma (ICC) | 22 (3.4%) | 101 (15.2%) | 38 (9.2%) | 42 (12.2%) | 119 (8.6%) | 161 (9.3%) |
| Poorly Specified Cancer Types (PSCT) | 414 (63.3%) | 431 (64.8%) | 299 (72.0%) | 170 (49.6%) | 974 (70.0%) | 1,144 (66.0%) |
| Rare Liver Cancers | 1 (0.2%) | 8 (1.2%) | 1 (0.2%) | 4 (1.2%) | 6 (0.4%) | 10 (0.6%) |
| **Cancer Stage a,b** | | | | | | |
| I stage | 19 (2.9%) | 21 (3.2%) | 7 (1.7%) | 22 (6.4%) | 25 (1.8%) | 47 (2.7%) |
| II stage | 34 (5.2%) | 41 (6.2%) | 13 (3.1%) | 51 (14.9%) | 37 (2.7%) | 88 (5.1%) |
| III stage | 86 (13.1%) | 85 (12.8%) | 27 (6.5%) | 59 (17.2%) | 139 (10.0%) | 198 (11.4%) |
| IV stage | 353 (54.0%) | 311 (46.8%) | 282 (68.0%) | 118 (34.4%) | 828 (59.5%) | 946 (54.6%) |
| Missing | 162 (24.8%) | 207 (31.1%) | 86 (20.7%) | 93 (27.1%) | 362 (26.0%) | 455 (26.2%) |
| **Basis of Diagnosis a,b** | | | | | | |
| Morphology | 227 (34.7%) | 255 (38.3%) | 144 (34.7%) | 189 (55.1%) | 437 (31.4%) | 626 (36.1%) |
| Other | 266 (40.7%) | 272 (40.9%) | 225 (54.2%) | 93 (27.1%) | 670 (48.2%) | 763 (44.0%) |

*(Continued)*

| | HCV Status* | | | Mortality Status as of December 2022 | | Overall |
|---|---|---|---|---|---|---|
| | Positive N = 654[1] (37.5%) | Negative N = 665[1] (38.4%) | Missing N = 415[1] (23.9%) | Alive N = 343[1] (19.8%) | Deceased N = 1,391[1] (80.2%) | N = 1,734[1] |
| Missing | 161 (24.6%) | 138 (20.8%) | 46 (11.1%) | 61 (17.8%) | 284 (20.4%) | 345 (19.9%) |
| **Degree of hepatic fibrosis***, *b,c,* | | | | | | |
| High | 472 (72.2%) | 0 (0.0%) | 0 (0.0%) | 118 (34.4%) | 354 (25.4%) | 472 (27.2%) |
| Gray zone | 12 (1.8%) | 0 (0.0%) | 0 (0.0%) | 2 (0.6%) | 10 (0.7%) | 12 (0.7%) |
| Low | 62 (9.5%) | 0 (0.0%) | 0 (0.0%) | 17 (5.0%) | 45 (3.2%) | 62 (3.6%) |
| Missing | 108 (16.5%) | 665 (100.0%) | 415 (100.0%) | 206 (60.1%) | 982 (70.6%) | 1,188 (68.5%) |

1 n (%); Median (IQR).

2 Hepatocellular Carcinoma (HCC): subset of liver cancer patients with GCR-assigned ICD-10 of C22.0; Intrahepatic cholangiocarcinoma (ICC): ICD-10 of C22.1; Rare types of liver cancers: ICD-10 C22.2-C22.4; Poorly Specified Cancer Types (PSCT): combined cases with ICD-10 C22.7 and ICD-10 C22.9.

a Significantly different among HCV statuses, as determined by Pearson's Chi-squared test or Welch Two Sample t-test.

b Significantly different among vital statuses, as determined by Pearson's Chi-squared test or Welch Two Sample t-test.

c Significance test by HCV status not applicable.

*HCV-negative status defined as: a) anti-HCV (-) or: b) anti-HCV (+) and HCV RNA (-)/HCVcAg (-) result. HCV-positive status defined as: HCV RNA (+)/HCVcAg (+) result regardless of anti-HCV result. Cases without HCV RNA or HCVcAg testing were defined as unknown and excluded from the analysis.

**Includes 3 cases with negative or missing anti-HCV results, but positive HCV RNA or HCVcAg results

***Liver fibrosis categories: low (Transient elastography (TE) score between F0-F2 or Fibrosis 4 (FIB4) score <1.45), high (TE score of F3 and F4, or FIB4 score >3.25), and gray zone (in cases with missing TE, those with a FIB4 score between 1.45 and 3.25). Missing status was given to cases with no present fibrosis evaluation record.

Out of 716 (91.2%) anti-HCV positive subjects tested for active infection, 654 (91.3%) tested positive for HCV RNA or HCVcAg (Fig 4).

Subjects that tested positive for active infection were significantly younger (median age of 59) compared to HCV negative subjects (median age of 67) at the time of cancer diagnosis (p-value < 0.001) (Table 1). The majority, 87.8%, of those that tested positive for active HCV infection were males (Table 1). 72.2% of HCV positive PLC cases assessed for liver fibrosis through the HCV Elimination Program were determined to have liver fibrosis based on transient elastography (TE) or Fib-4 score during their pre-treatment workup (Table 1).

Overall, 495 (75.7%) HCV positive PLC patients identified in the GCR received treatment for HCV infection under the national HCV Elimination Program (Table 1 and Fig 4), with 355 (71.7%) cases enrolling into the HCV treatment program more than one year before their PLC diagnosis and 67 (13.5%) initiating HCV treatment after their PLC diagnosis (Fig 5).

Of the subjects who completed at least one round of HCV treatment, 75.0% were assessed for SVR achievement (Fig 4). 90.3% of subjects successfully achieved SVR post-treatment (Fig 4).

The majority of PLC cases were missing data on HIV and HBV co-infections, with only 17.2% reporting information on HIV and only 30.4% having HBV status information. Among the HCV-infected subgroup, 1 (0.3%) person reported having HIV and 13 (2.5%) people tested positive for HBsAg (Table 1).

## Discussion

This study identified a total of 1734 adult cases of primary liver cancer registered in the GCR between 2015 and 2022. Our findings revealed differences in characteristics among age and sex subgroups, with most cases occurring in males above age 60. This pattern aligns with previously published studies that indicate a mean age of diagnosis between 63

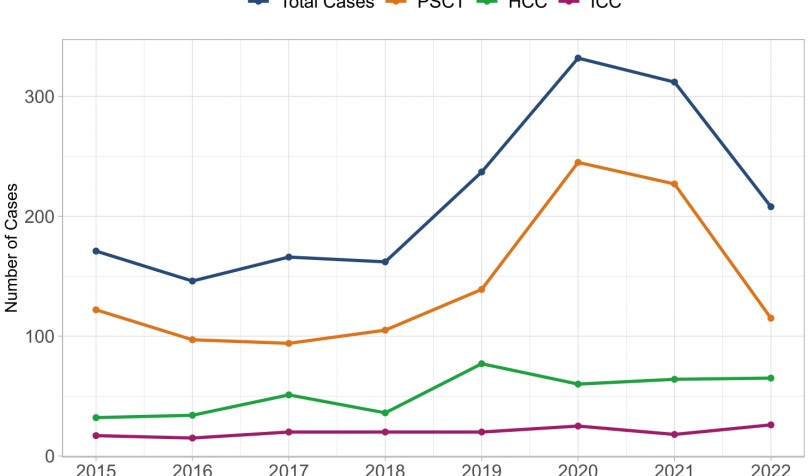

**Fig 1. Annual number of primary liver cancer (PLC) Cases in Georgia, 2015-2022.** Data from the Georgian Cancer Registry presented as a line graph, illustrating trends in registered adult PLC cases over the 8-year period.

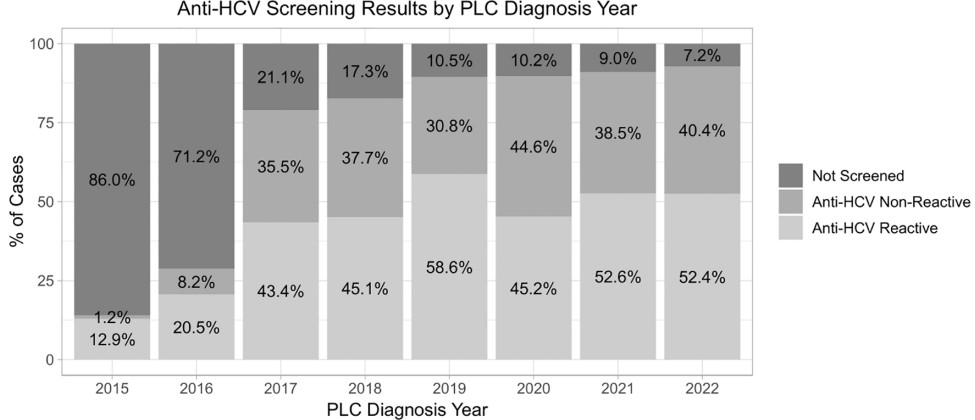

**Fig 2. HCV Screening rate among primary liver cancer (PLC) cases in Georgia, 2015-2022.** The bar graph illustrates the annual percentage of newly diagnosed PLC patients screened at least once for anti-HCV, showing the increase in screening coverage over time.

and 65 years in Europe and North America [5]. Older age is considered a risk factor for PLC due to prolonged exposure to chronic HCV infection and consequently, an increased risk of advanced liver damage [5,14,23]. The majority of our retrospective cohort was deceased by the end of 2022, highlighting the importance of early diagnosis and treatment.

Our findings suggest that males are more likely to be investigated for and diagnosed with HCV infection compared to females and have better follow-up for linkage to care. Males also had a higher likelihood of being diagnosed with hepatocellular carcinoma (HCC) compared to other subtypes of liver cancers. Females tended to be older than males at the time of cancer diagnosis, and were mostly classified with poorly specified liver cancer types, suggesting a potential sex disparity in PLC diagnosis. The higher prevalence of primary liver cancer (PLC) in males may likely be explained in part by the greater prevalence of HCV infection among males in the Georgian population [9]. This observation is consistent with

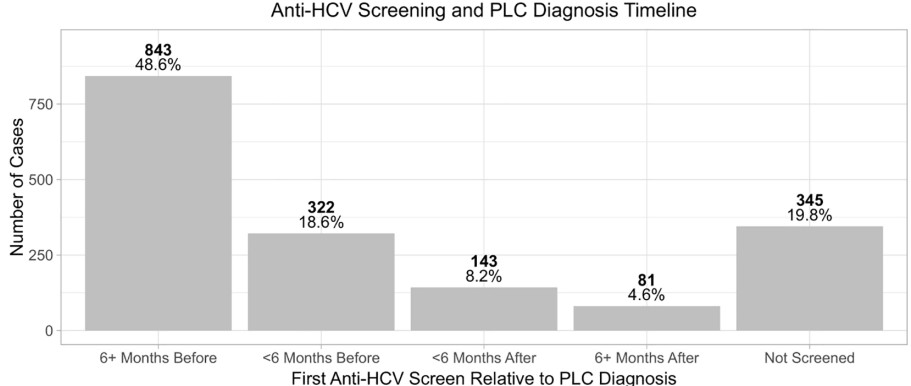

**Fig 3. Timing of anti-HCV testing relative to primary liver cancer (PLC) diagnosis among registered PLC cases in Georgia, 2015-2022.** The bar graph illustrates the proportion of first recorded anti-HCV test results in relation to the date of PLC diagnoses, grouped in 6-month intervals.

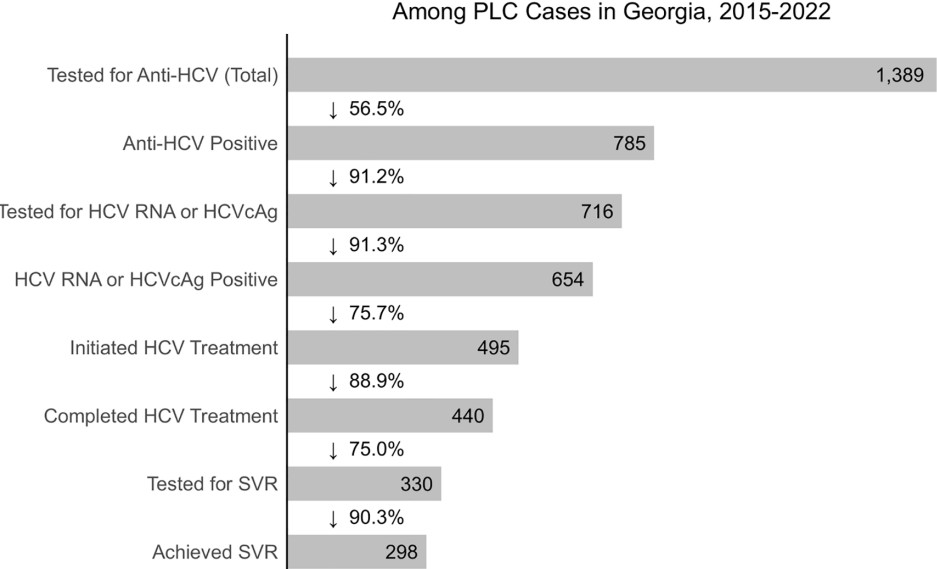

**Fig 4. Care Cascade for Hepatitis C Testing and Treatment among PLC Cases in Georgia, 2015-2022.** The bar graph shows the cumulative absolute numbers and proportion of the PLC cases assessed and treated for HCV infection. SVR achievement was defined for only those who completed at least one round of treatment and was based on patients' latest round of treatment.

other studies that have found a higher prevalence of HCC among men, particularly in areas where the distribution of HCV infection is higher among the male population [5].

Further, only 419 (24.2%) cases were classified as hepatocellular carcinoma (HCC), even though HCC is estimated to account for 75–85% of global PLC cases [1]. Considering that only a small proportion of cases (36.1%) had morphology-based cancer diagnoses, our findings support the claim that HCC diagnoses are underreported in the country. There is an overall increase in the number of both PLC and HCC cases throughout the study period, with a peak number of cases reported in 2020. This general increase can potentially be attributed to the implemented changes in

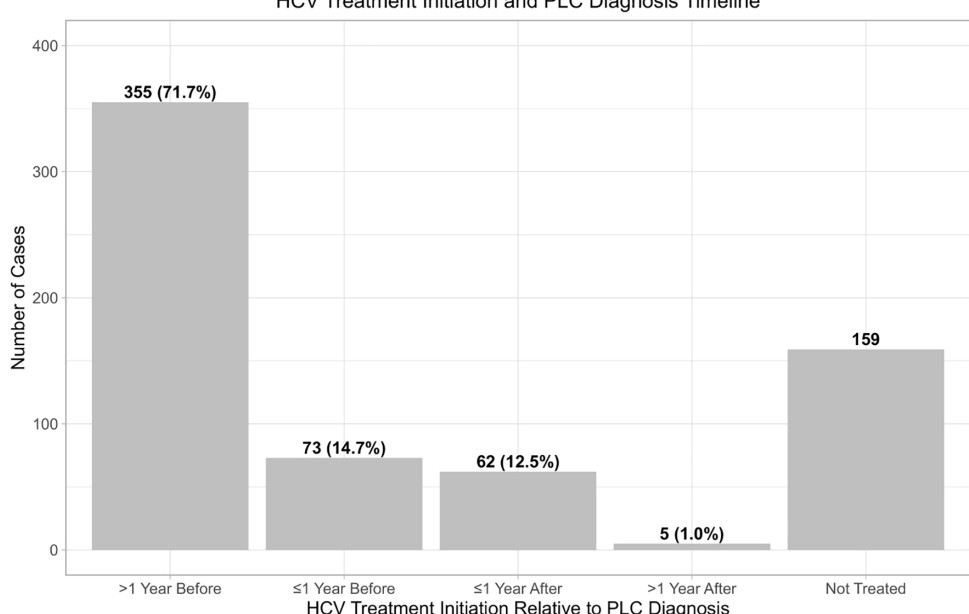

**Fig 5. Timing of the HCV treatment initiation relative to primary liver cancer (PLC) diagnosis among registered PLC cases in Georgia, 2015-2022.** The bar graph illustrates the proportion of initiated earliest round of treatment in relation to the date of PLC diagnoses, grouped in 1-year intervals.

case reporting into the cancer registry [16]: In 2020, the cancer reporting process was updated to include those with cancer-based causes of death in issued death certificates as cases. This update serves as a potential explanation for the peak in reported cases in 2020. Moreover, the relative decrease in 2022 is explained by the incomplete data, as at the time of data extraction, cancer diagnoses based on issued death certificates in 2022 has not been fully incorporated into the GCR. Due to these changes and updated regulations aimed at improving the cancer reporting in Georgia over the years, it is difficult to assess the potential role the HCV Elimination Program played in liver cancer incidence in the country over the study period. However, the findings of previous research [24], coupled with the fact that the most severe HCV cases were identified and treated within the program in its earliest years, highlight the potential of decrease in the number of HCV-attributable liver cancer cases over time due to the increase in HCV treatment. It is important to continue this analysis with future data to better quantify the impact of the HCV Elimination Program on liver cancer incidence in Georgia.

Most liver cancer patients in our cohort were investigated for Hepatitis C infection. Screening rates for HCV infection have steadily increased over time, culminating in 92.8% of cancer cases having been tested for HCV antibodies in 2022. This indicates the effectiveness of the national HCV Elimination Program in providing greater access to testing. Lower HCV testing rates among PLC cases in the beginning of the program might be explained by fact that, although universal screening was available to all citizens, initially only individuals with established cirrhosis were eligible for treatment provided at four specialized sites [10,25]. In 2016, the program adopted a "treatment-for-all" approach, expanding eligibility to all HCV-infected individuals and increasing the number of treatment sites thought the country, significantly improving access to diagnostic and treatment services [26,27]. Moreover, the removal of patient co-payment for treatment workup and monitoring in 2019 made the program services even more accessible [26]. Our results illustrate the impact of this expansion, as we see more people being investigated for HCV and enrolled in treatment over time [13,27].

In this cohort, cancer cases with active HCV infection were significantly younger (median age 59) compared to HCV-negative subjects (median age 67) at the time of cancer diagnosis (p-value < 0.001). This observation could be

interpreted as evidence of the success of the HCV Elimination Program in referring patients to oncologists during pre-treatment workups. Additionally, research shows that HCV infection is associated with liver cancer development at earlier ages comparing to other risk factors, potentially also explaining this finding [28].

Co-infection with hepatitis B was found in 2.5% of tested HCV-infected individuals, which is similar to the national average of hepatitis B prevalence of 2.7%. [29]. As expected, given the low prevalence of HIV infection among the general population (0.4%) [30], with the infection mainly concentrated among the high-risk groups (15.3% among men who have sex with men (MSM) [31]), the number of patients who self-reported having HIV infection (n = 1, 0.1%) is comparable to the prevalence among the general population, 0.4%. Sociocultural norms and trends such as these contribute to a stigma that discourages patients from reporting positive HIV statuses in Georgia. Because our data collection did not include HIV-specific focus, the association between HIV and liver cancer within the Georgian population remains to be further investigated.

Approximately 76% of HCV-infected PLC cases received HCV treatment through the national HCV Elimination Program between 2015 and 2022, either before or after their cancer diagnosis, highlighting the effectiveness of the program in linking individuals to HCV treatment and care. However, we see a lower rate of sustained virological response (SVR) in HCV-infected PLC individuals (90.3%) compared to the national average after at least one round of treatment (99.0%) [13]. We hypothesize this might be partially accounted for by variations in SVR rates among different treatment regimens implemented by the program over time: SVR rates ranged from 82.3% with Sofosbuvir-only therapy used from the initiation until April 2016, 98.2% with SOF/LED regimens subsequently used until December 2018, to 98.6% with SOF/VEL regimens used after December 2018 [13]. Alternatively, it is also well-documented that structural and immunological changes of hepatocytes due to advanced fibrosis [32] and the formation of hepatocellular carcinoma contribute to relatively lower observed SVR rates among HCV-infected PLC cases receiving DAA treatment [33–35].

The key finding of our analysis showed nearly 72% of HCV-infected individuals with PLC had received HCV treatment under the national HCV Elimination Program more than one year before being diagnosed with primary liver cancer. This highlights the importance of implementing systematic routine follow-up care for patients after HCV treatment, especially those with advanced liver disease. Currently, under the Elimination Program, treated patients are not followed after treatment is completed. Initiating routine post-treatment monitoring, including imaging and testing for liver cancer biomarkers within the national HCV Elimination Program could enhance the diagnosis and outcomes of primary liver cancer in individuals who have undergone HCV treatment.

## Limitations

The national surveillance systems used in this analysis were introduced at the beginning of the study period. There were multiple iterations of data collection processes before these systems became fully functional, limiting the available data from earlier operational years. The results of this analysis are undermined by a lack of information on potential confounding factors of PLC, including metabolic markers, alcohol use, substance use, co-infections and environmental factors. These variables are either significantly underreported, like alcohol and substance use and HIV co-infection, or challenging to estimate, such as exposure to environmental pollutants. Consequently, we are unable to provide a comprehensive look at the factors driving primary liver cancer development in the Georgian population. Given that almost half of the PLC cases in our cohort tested negative for HCV infection, the newly launched National Hepatitis B Management Program in the country could aid screening for HBV infection and contribute to a better understanding of this risk factor for primary liver cancer beyond hepatitis C infection.

## Conclusion

This analysis provides a comprehensive description of primary liver cancer in Georgia in the context of the hepatitis C elimination program. Our findings confirm a substantial burden of HCV infection among primary liver cancer cases

and demonstrate the success of the national program in expanding Hepatitis C screening and treatment coverage. At the same time, the persistently high proportion of late-stage cancer diagnoses and mortality highlights ongoing gaps in early cancer detection. The findings demonstrated that the majority of HCV-infected individuals with PLC had received antiviral treatment prior to cancer diagnosis, highlighting a need to integrate long-term post-treatment surveillance, particularly targeted at high-risk individuals, into the national elimination framework to support earlier detection of liver cancer and improve outcomes, maximizing the long-term public health impact of Georgia's hepatitis C elimination efforts.

## Acknowledgments

The authors would like to acknowledge the contributions of their colleagues from the Centers for Disease Control and Prevention (CDC), specifically Dr. Paige A. Armstrong for her role in the study's conception and design, and Shaun Shadaker, MPH for his invaluable support in its implementation.

## Author contributions

**Conceptualization:** Anna Khoperia, Geoff A Beckett, Maia Tsereteli, Maia Alkhazashvili, Amiran Gamkrelidze, Francisco Averhoff.

**Data curation:** Anna Khoperia, Nazibrola Chitadze, Konstantine Kazanjan, Vladimer Getia.

**Formal analysis:** Anna Khoperia, Susan H Gawel, Jenna Bedrava, Maia Butsashvili, Alan L Landay, Sophia Surguladze.

**Funding acquisition:** Anna Khoperia.

**Methodology:** Anna Khoperia, Susan H Gawel, Francisco Averhoff.

**Project administration:** Anna Khoperia, Susan H Gawel.

**Resources:** Anna Khoperia, Gavin A Cloherty, Francisco Averhoff.

**Supervision:** Susan H Gawel, Nazibrola Chitadze, Maia Butsashvili, Francisco Averhoff.

**Validation:** Susan H Gawel, Jenna Bedrava, Alan L Landay, Sophia Surguladze, Gavin A Cloherty, Francisco Averhoff.

**Visualization:** Jenna Bedrava.

**Writing – original draft:** Anna Khoperia.

**Writing – review & editing:** Susan H Gawel, Jenna Bedrava, Nazibrola Chitadze, Maia Butsashvili, Alan L Landay, Geoff A Beckett, Sophia Surguladze, Konstantine Kazanjan, Vladimer Getia, Maia Tsereteli, Maia Alkhazashvili, Gavin A Cloherty, Mamuka Zakalashvili, Amiran Gamkrelidze, Francisco Averhoff.

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
