## [Decision Letter · Decision Letter 0]

1 Feb 2026

Dear Dr. Khoperia,

We look forward to receiving your revised manuscript.

Kind regards,

Jason T. Blackard, PhD

Academic Editor

PLOS One

Journal Requirements:

“This publication has been produced during the corresponding author’s fellowship from the joint TEPHINET/APDC (Abbott Pandemic Defense Coalition) Field Epidemiology Training Program. This publication was supported by a Memorandum of Agreement funded by Abbott. Its contents are solely the responsibility of the authors and do not necessarily represent the official views of Abbott, The Task Force for Global Health, Inc. or TEPHINET”

Additional Editor Comments:

This is a descriptive analysis of HCV infection and treatment conducted in the Republic of Georgia.

How likely is it that primary liver cancer cases are never reported?  How comprehensive is the cancer registry system?

The two reviewers raise multiple issues that should be addressed appropriately in a revised submission.

Reviewers' comments:

Reviewer's Responses to Questions

**Comments to the Author**

1. Is the manuscript technically sound, and do the data support the conclusions?

Reviewer #1: Partly

Reviewer #2: Yes

2. Has the statistical analysis been performed appropriately and rigorously?

Reviewer #1: Yes

Reviewer #2: Yes

3. Have the authors made all data underlying the findings in their manuscript fully available?

Reviewer #1: Yes

Reviewer #2: Yes

4. Is the manuscript presented in an intelligible fashion and written in standard English?

Reviewer #1: Yes

Reviewer #2: Yes

Reviewer #1: Manuscript Number: PONE-D-25-61009

Title: Primary liver cancer in Georgia: Seven years’ experience following the launch of

the Hepatitis C Elimination Program, 2015-2022

I appreciate the opportunity to review this report on the impact of HCV elimination on primary liver cancer and commend the authors for studying this important topic.

The authors present an original, retrospective, descriptive analysis of HCV infection, treatment, and cure in patients diagnosed with primary liver cancer in Georgia over an 8-year period following implementation of the national HCV Elimination Program. The stated aim is to describe and evaluate the prevalence of HCV infection among liver cancer patients and assess the HCV Elimination Program outcomes within this population. They utilize the Georgian Cancer Registry to identify cases and merge data from the national HCV treatment registries. The results of this study effectively highlight the success of the Georgian HCV elimination program in increasing the rates of HCV screening and treatment and emphasize the need for continued HCC surveillance following diagnosis and treatment of HCV, especially in those with advanced fibrosis/cirrhosis.

Introduction

The introduction is clear and concise providing relevant background information regarding the association of HCV and hepatocellular carcinoma and the epidemiology of HCV in Georgia. The Georgian HCV elimination program and the relevant national registries/databases are appropriately introduced to give context to the current manuscript. However, the study lacks a clear scientific aim and focus

Methods

The methods are appropriate. Statistical methods are appropriately described; the one exception is that there is no mention of adjusting statistical methods for multiple comparisons (multiplicity).

The term Poorly Specified Cancer Types (PSCT) is mentioned in Table 1 but does not match with the term in the methods. These should match to minimize any confusion. In addition, 63% of cancer cases falling in this category is too high and would limit any meaningful data interpretation

On the same note, Table 1 suggests 161 missing under “Basis of Diagnosis” which also limits data interpretation

127 HCV patients are missing Hep B status diagnosis thus limiting the data. Since Hep B is an independent risk factor for HCC, this data is valuable

Patients were evaluated for liver fibrosis using FIB-4 as well as transient elastography. It is not clear which modality was finally used to label the fibrosis stage for the final analyses. In addition, 108 cases are missing fibrosis estimate thus limiting data interpretation

Results

Liver cancer stage is reported as stage I-IV. It is unclear what staging systems were used but one would assume the traditional TMN staging systems. This is difficult to interpret in this study because the majority of cancers reported were “poorly specified cancer types” and the remainder were a mixture of HCC and cholangiocarcinoma. TMN staging is generally not used in treatment/prognosis of HCC, rather the Barcelona-Clinic Liver Cancer (BCLC) staging system is the preferred system. I suspect this information is not available with the methodology employed in this study and should be discussed as another limitation.

Table 1: “level of liver damage” would be more accurately described as “degree of hepatic fibrosis”

p. 16, line 287: consider changing “liver damage” to “liver fibrosis based on transient elastography (TE) or Fib-4 score”

Discussion

The authors appropriately highlighted the apparent sex disparity in HCV screening. The authors note lower SVR rates of 90.3% in individuals with primary liver cancer compared to a national average of 99%; they offer evolving antiviral regimens and HCC itself as possible explanations for the lower SVR rate. I would also suggest that since vast majority of these patients has stage 3 or 4 fibrosis, this would also result in lower SVR rates. Other significant limitations of this study including a lack of accounting for well established risk factors for HCC (alcohol, cirrhosis, HBV, etc) are appropriately acknowledged.

Regarding the key findings of the study that HCC is occurring post SVR, this is not a novel finding. AASLD guidelines recommend ongoing HCC surveillance/screening in patients post SVR with cirrhosis. Following is except from AASLD guidelines:

Available data demonstrate patients with HCV cirrhosis remain at an increased HCC risk for up to 10 years after SVR, so surveillance should be continued indefinitely unless future data demonstrate sufficiently reduced HCC incidence. Patients with HCV and NAFLD without cirrhosis, particularly those with advanced fibrosis, would benefit from risk stratification tools to identify those at highest risk to whom surveillance could be targeted in the future. In the interim, surveillance may be considered in select patients with advanced fibrosis on a case-by-case basis, particularly for those in whom there is clinical suspicion for understaging of fibrosis by noninvasive markers or biopsy.

Since vast majority of patients in this study with HCV and HCC had advanced fibrosis/cirrhosis, these would fall under such guidelines. Missing data in the study could account for the rest of the HCC patients that developed HCC post treatment

References:

El-Serag HB, Kanwal F, Feng Z, Marrero JA, Khaderi S, Singal AG. Texas Hepatocellular Carcinoma C. Risk Factors for Cirrhosis in Contemporary Hepatology Practices-Findings From the Texas Hepatocellular Carcinoma Consortium Cohort. Gastroenterology. 2020;159:376–377

Ioannou GN, Beste LA, Green PK, Singal AG, Tapper EB, Waljee AK, et al. Increased Risk for Hepatocellular Carcinoma Persists Up to 10 Years After HCV Eradication in Patients With Baseline Cirrhosis or High FIB-4 Scores. Gastroenterology. 2019;157:1264–1278 e1264

Singal, Amit G.; Llovet, Josep M.; Yarchoan, Mark; Mehta, Neil6; Heimbach, Julie K.; Dawson, Laura A.; Jou, Janice H.; Kulik, Laura M.; Agopian, Vatche G.; Marrero, Jorge A.; Mendiratta-Lala, Mishal; Brown, Daniel B.; Rilling, William S.; Goyal, Lipika; Wei, Alice C.; Taddei, Tamar H. Hepatology 78(6):p 1922-1965, December 2023.

Reviewer #2: Thank you for asking me to review this manuscript looking at the rate of PLC in the Georgian HCV elimination program. Unsurprisingly, HCC was noted and associated with older age and advanced chronic liver disease. Points of clarity/questions include:

1. Cancer (PLC etc) is staged – what staging system is being referred to? A global TNM stage, BCLC for HCC? Could this please be clarified?

2. The authors state for diagnosis: The basis of liver cancer subtype diagnosis was divided into two categories: tissue-based diagnosis (including histology, cytology, immunohistochemistry and immunophenotyping) and diagnosis based on other diagnostic criteria (including clinical, instrumental, radiological and specific cancer markers). Could the authors clarify (a) what is meant by instrumental? , and (b) radiological – was there any standardization for example with PLC such as using the LIRADS system and (c) what cancer markers? Are they referring too AFP, Ca19-9 ?

3. Its is unclear what is meant by cancer “screening” – was this based on ultrasound and AFP primarily? Was PLC screening post hep C diagnosis with advanced chronic liver disease performed 6 monthly? The authors suggest this is not routinely performed but cancers were diagnosed, mostly at an advanced stage – is this a consequence of no routine PLC screening performed?

4. Perhaps the authors could provide some brief context to the access and availability of services to assess and mange/offer therapeutic interventions for PLC and other liver cancers in Georgia.

5. Does the database, apart from HBV and HIV co-infection, offer any insights into co-factors such as alcohol use, MASLD etc?

.

Reviewer #1: No

Reviewer #2: No

---

## [Author Response · Author response to Decision Letter 1]

18 Mar 2026

Editor Comments to the Author

Author response: We thank the editor for the thoughtful feedback and constructive suggestions.

The manuscript has been reviewed and revised to ensure compliance with PLOS ONE formatting and style requirements, including file naming, formatting, and section structure.

Author response: We thank the editor for the thoughtful feedback.

The manuscript has been reviewed and revised to ensure compliance with PLOS ONE requirements regarding funding-related information and any funding-related text was removed from the manuscript.

“This publication has been produced during the corresponding author’s fellowship from the joint TEPHINET/APDC (Abbott Pandemic Defense Coalition) Field Epidemiology Training Program. This publication was supported by a Memorandum of Agreement funded by Abbott. Its contents are solely the responsibility of the authors and do not necessarily represent the official views of Abbott, The Task Force for Global Health, Inc. or TEPHINET”

Author response: Thank you for this comment. We have updated the financial disclosure statement to reflect the funders’ role in the study:

“This publication has been produced during the corresponding author’s fellowship from the joint TEPHINET/APDC (Abbott Pandemic Defense Coalition) Field Epidemiology Training Program. This publication was supported by a Memorandum of Agreement funded by Abbott. Its contents are solely the responsibility of the authors and do not necessarily represent the official views of Abbott, The Task Force for Global Health, Inc. or TEPHINET. The co-authors from Abbott participated as employees and shareholders of Abbott.”

b) If there are no restrictions, please upload the minimal anonymized data set necessary to replicate your study findings to a stable, public repository and provide us with the relevant URLs, DOIs, or accession numbers. For a list of recommended repositories, please see https://journals.plos.org/plosone/s/recommended-repositories. You also have the option of uploading the data as Supporting Information files, but we would recommend depositing data directly to a data repository if possible.

Author response: Thank you for this important comment. This study relied on linked, individual-level data from multiple national surveillance systems in Georgia, including the Georgian Cancer Registry, the national Vital Registration System, and the Hepatitis C Elimination Program databases. These data are government-owned and governed by national data protection legislation.

Although identifiers were encrypted prior to analysis, the dataset contains detailed clinical, temporal, and mortality information derived from multiple national registries. The ethical approval granted by the Institutional Review Board of the National Center for Disease Control and Public Health (IRB #2023-043) authorized the use of these data for the purposes of this approved research but does not permit public release of individual-level data. In addition, national regulations prohibit public dissemination of registry-level data derived from state surveillance systems. Therefore, the data cannot be made publicly available. Requests for access may be considered by the National Center for Disease Control and Public Health and the Ministry of Health in accordance with national data governance procedures and subject to formal review and data use agreements. For more information, please contact the Institutional Review Board of the National Center for Disease Control and Public Health at irb.ncdc@ncdc.ge

Author response: Thank you for this comment. We have updated the methods section of the manuscript to include the ethics statement. (Page 6, line 116-117).

Additional Editor Comments:

This is a descriptive analysis of HCV infection and treatment conducted in the Republic of Georgia. How likely is it that primary liver cancer cases are never reported? How comprehensive is the cancer registry system?

Author response: We thank you for raising such a good comment.

Since the development of Cancer Registry in 2015, there have been several regulatory changes mandating and improving the quality of the cancer case reporting in Georgia, making it somewhat difficult to assess the extent of lost liver cancer cases left beyond the systematic reporting within the study period.

However, overall, the Georgian Cancer registry was built and improved over the years to increase the coverage and accuracy of case-based cancer surveillance in the country. According to the regulation, any medical service provider is obliged to report the case to the Cancer Registry. This applies to all hospitals, clinics, diagnostic and treatment and radiation centers. Medical institutions are expected to provide data to the cancer registry on at least 95% of the cases diagnosed or treated at their facilities [1], especially ever since it became tied to state-allocated cancer treatment funding in 2023.

The reporting was improved by launching the Unified Electronic System for Cancer Data Collection to start collecting data electronically and combining information on cancer screening, diagnosis and treatment in 2019 [2], and updating the reporting process to include those with cancer-based causes of death in issued death certificates as cases in 2020, as stated within the discussion section.

These actions resulted in doubling of the registered incidence of malignant neoplasms after the introduction of the Cancer registry [3]: In 2014, only 4,200 new cases were registered, while since 2015 and the implementation of the GCR, more than 10,000 cancer new cases have been detected annually [2,3]. In 2015, according to the GCR data, there were registered 10,506 new cases of malignant neoplasms [4], and it reached 11.900 by 2024 [5]

1. National Center for Disease Control and Public Health (NCDC). National cancer control strategy. Tbilisi: Ministry of Labour, Health and Social Affairs of Georgia; 2014. Available from: https://test.ncdc.ge/Handlers/GetFile.ashx?ID=c3e64991-853c-4ab5-9cc0-2e04ac15193b

2. International Association of National Public Health Institutes (IANPHI). Georgia’s population-based cancer registry: a success story of surveillance [Internet]. Saint-Maurice: IANPHI; 2022. Available from: https://ianphi.org/news/2022/georgia-cancer-registry.html

3. National Center for Disease Control and Public Health. Medical statistics: Georgia brief 2020 [Internet]. Available from: https://test.ncdc.ge/Handlers/GetFile.ashx?ID=0f328262-15c1-41d8-b01e-daa9779fe87d

4. Abesadze N. History of cancer registration in Georgia. Cauc J Health Sci Public Health. 2018;2(2):88–92

5. National Statistics Office of Georgia (Geostat). Statistical yearbook of Georgia 2024 [Internet]. Tbilisi: Geostat; 2024. Available from: https://www.geostat.ge/media/67769/Yearbook_2024.pdf

Reviewer 1 Comments to the Author

Introduction

The introduction is clear and concise providing relevant background information regarding the association of HCV and hepatocellular carcinoma and the epidemiology of HCV in Georgia. The Georgian HCV elimination program and the relevant national registries/databases are appropriately introduced to give context to the current manuscript. However, the study lacks a clear scientific aim and focus

Author response: We thank the reviewer for the thoughtful feedback. The introduction section was revised and paragraph added to clarify the aims and focus of the study (Page 5, lines 93-104):

“Despite Georgia’s substantial progress toward HCV elimination, the characteristics of primary liver cancer cases and their relationship to HCV infection and treatment in the era of national elimination remain incompletely described. In particular, there is limited evidence on the burden of active and treated HCV infection among PLC patients and the timing of antiviral treatment relative to cancer diagnosis at the population level. Using linked data from the Georgian Cancer Registry, the national HCV screening and treatment databases, and the national vital registration system, this study aimed to characterize the epidemiologic and clinical profile of PLC cases in Georgia from 2015–2022, determine the prevalence of active and treated HCV infection among them and evaluate patterns of HCV testing, treatment uptake, and timing of treatment in relation to PLC diagnosis to provide important baseline evidence to inform liver cancer surveillance strategies and guide long-term management of patients following HCV treatment in the context of national elimination efforts.”

The methods

The methods are appropriate. Statistical methods are appropriately described; the one exception is that there is no mention of adjusting statistical methods for multiple comparisons (multiplicity).

The term Poorly Specified Cancer Types (PSCT) is mentioned in Table 1 but does not match with the term in the methods. These should match to minimize any confusion. In addition, 63% of cancer cases falling in this category is too high and would limit any meaningful data interpretation

On the same note, Table 1 suggests 161 missing under “Basis of Diagnosis” which also limits data interpretation

127 HCV patients are missing Hep B status diagnosis thus limiting the data. Since Hep B is an independent risk factor for HCC, this data is valuable.

Patients were evaluated for liver fibrosis using FIB-4 as well as transient elastography. It is not clear which modality was finally used to label the fibrosis stage for the final analyses. In addition, 108 cases are missing fibrosis estimate thus limiting data interpretation.

Author response: Thank you for your comment. We agree that multiplicity adjustments are important when conducting series of formal hypothesis tests for the purpose of drawing inferential conclusions. However, Table 1 in our manuscript is presented for descriptive purposes only, intended to summarize characteristics of the study population across groups. It is standard practice to report unadjusted p-values in Table 1s and therefore, we have kept the unadjusted descriptive comparisons in our manuscript.

Text was updated to ensure the term Poorly Specified Cancer Types (PSCT) is mentioned in Table 1 has been kept consistent.

As stated in the limitation section, the high percentages of missing data reflect real-world clinical documentation challenges in the early years of the registry, which is a key finding in itself regarding the need for registry improvement. These gaps in the data can actually provide a benchmark for the reference in this area of research in the future.

To label the fibrosis stage for the final analyses, the combination of both FIB-4 as well as transient elastography results were used. For cases having both results available in the dataset, elastography score was given priority over FIB4 score, classifying the cases into low, high and gray zone liver fibrosis categories. Missing status was given to cases if with no records of fibrosis evaluation was present.

Results

Liver cancer stage is reported as stage I-IV. It is unclear what staging systems were used but one would assume the traditional TMN staging systems. This is difficult to interpret in this study because the majority of cancers reported were “poorly specified cancer types” and the remainder were a mixture of HCC and cholangiocarcinoma. TMN staging is generally not used in treatment/prognosis of HCC, rather the Barcelona-Clinic Liver Cancer (BCLC) staging system is the preferred system. I suspect this information is not available with the methodology employed in this study and should be discussed as another limitation.

Table 1: “level of liver damage” would be more accurately described as “degree of hepatic fibrosis”

p. 16, line 287: consider changing “liver damage” to “liver fibrosis based on transient elastography (TE) or Fib-4 score”.

Author response: Thank you for your important remark. Within the Georgian Cancer Registry, all reported cases are staged based on the TNM classification system. The following sentence was added into the methodology section for clarification: (p. 6, lines 121-125)

„For each registered case, the GCR collects the data on confirmed cancer cases using 3rd revision of the International Classification of Disease for Oncology (ICD-O-3) coding, stage (according to Tumor Nodes Metastases (TNM) classification system), results of laboratory tests used in diagnosis, assigned treatment, and outcomes [16]. “

Table 1: “level of liver damage” description changed to “degree of hepatic fibrosis”; (p.13)

p.16, line 287 changed as suggested (line 267-268).

Discussion

The authors appropriately highlighted the apparent sex disparity in HCV screening. The authors note lower SVR rates of 90.3% in individuals with primary liver cancer compared to a national average of 99%; they offer evolving antiviral regimens and HCC itself as possible explanations for the lower SVR rate. I would also suggest that since vast majority of these patients has stage 3 or 4 fibrosis, this would also result in lower SVR rates. Other significant limitations of this study including a lack of accounting for well-established risk factors for HCC (alcohol, cirrhosis, HBV, etc) are appropriately acknowledged.

Regarding the key findings of the study that HCC is occurring post SVR, this is not a novel finding. AASLD guidelines recommend ongoing HCC surveillance/screening in patients post SVR with cirrhosis. Following is except from AASLD guidelines:

Available data demonstrate patients with HCV cirrhosis remain at an increased HCC risk for up to 10 years after SVR, so surveillance should be continued indefinitely unless future data demonstrate sufficiently reduced HCC incidenc

---

## [Decision Letter · Decision Letter 1]

7 Apr 2026

Title: Primary liver cancer in Georgia: Seven years’ experience following the launch of the Hepatitis C Elimination Program, 2015-2022

PONE-D-25-61009R1

Dear Dr. Khoperia,

We’re pleased to inform you that your manuscript has been judged scientifically suitable for publication and will be formally accepted for publication once it meets all outstanding technical requirements.

Kind regards,

Jason T. Blackard, PhD

Academic Editor

PLOS One

Additional Editor Comments (optional):

None

Reviewers' comments:

Reviewer's Responses to Questions

**Comments to the Author**

Reviewer #1: All comments have been addressed

Reviewer #2: All comments have been addressed

2. Is the manuscript technically sound, and do the data support the conclusions?

Reviewer #1: Yes

Reviewer #2: Yes

3. Has the statistical analysis been performed appropriately and rigorously?

Reviewer #1: Yes

Reviewer #2: N/A

4. Have the authors made all data underlying the findings in their manuscript fully available?

Reviewer #1: Yes

Reviewer #2: Yes

5. Is the manuscript presented in an intelligible fashion and written in standard English?

Reviewer #1: Yes

Reviewer #2: Yes

Reviewer #1: Thank you for detailed responses to all the queries and suggestions. I understand some data limitations but these have now been highlighted and rationale has been provided

Reviewer #2: (No Response)

.

Reviewer #1: No

Reviewer #2: No

---

## [Editor Report · Acceptance letter]

PONE-D-25-61009R1

PLOS One

Dear Dr. Khoperia,

I'm pleased to inform you that your manuscript has been deemed suitable for publication in PLOS One. Congratulations! Your manuscript is now being handed over to our production team.

Kind regards,

on behalf of

Dr. Jason T. Blackard

Academic Editor

PLOS One